# Model-Agnostic Graph Dataset Compression with the Tree Mover's Distance

**Mika Sarkin Jain**[1] **Stefanie Jegelka**[2] **Ishani Karmarkar**[1] **Luana Ruiz**[3] **Ellen Vitercik**[1]

## Abstract

Graph neural networks have demonstrated remarkable success across a variety of domains. However, the acquisition and management of large-scale graph datasets poses several challenges. Acquiring graph-level labels can be prohibitively costly, especially for applications in the biosciences and combinatorial optimization. Storage and privacy constraints can pose additional challenges. In this work, we propose an approach for data subset selection for graph datasets, which downsamples graphs and nodes based on the Tree Mover's Distance. We provide new efficient methods for computing the TMD in our setting; empirical results showing our approach outperforms other node and graph sampling methods; and theoretical results bounding the decrease in accuracy caused by training on the downsampled graphs. Surprisingly, we find that with our method, we can subsample down to 1% of the number of graphs and 10% of the number of nodes on some datasets, with minimal degradation in model accuracy.

## 1. Introduction

Graph neural networks (GNNs) are a popular architecture for learning over graph-structured data. Despite their widespread popularity and demonstrated success (Bongini et al., 2022; Zhou et al., 2020; Peng et al., 2021; Fan et al., 2019; Peng et al., 2021; Zhang & Chen, 2018), training GNNs remains challenging because the datasets can be massive, both in terms of the size (number of nodes) of each graph, as well as the number of graphs in the dataset.

Large-scale datasets can pose many challenges to training GNNs. For instance, training may be computationally intractable (Hashemi et al., 2024; Yan et al., 2020; Zeng et al., 2021). First of all, obtaining supervision signals can be

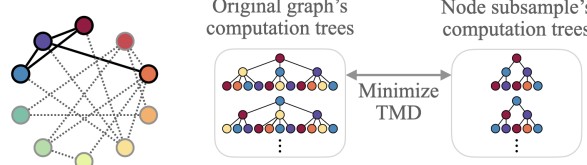

*Figure 1.* Node subsampling. Given node budget $k$, we want to find a subgraph of $k$ nodes with minimum TMD from the original.

expensive on large graphs or on datasets consisting of large graphs. This is the case, for example, when GNNs are used to model molecular properties for computational drug discovery (Igarashi et al., 2024; Shen et al., 2020; Jiang et al., 2021). Likewise, storing and loading massive graphs into memory can be slow and/or expensive (Huang et al., 2024; Hamilton et al., 2017; Da San Martino et al., 2012) or may raise privacy concerns (Wu et al., 2022). GNNs have also been employed to circumvent expensive simulation, e.g., in molecular dynamics (Gasteiger et al., 2021; Park et al., 2021; Li et al., 2022), where the cost of labeling larger instances grows quickly with graph size. Similarly, GNNs have been applied for disease prediction (Sun et al., 2020; Zheng et al., 2022), in which case labeling a single graph may require expensive clinical or diagnostic procedures. Additionally, there is growing interest in using GNNs to solve challenging or costly optimization problems, in which case the runtime to collect training labels grows with the graph size (e.g., Bengio et al., 2021; Cappart et al., 2023; Vesselinova et al., 2020).

A natural approach to tackle these multifaceted challenges is *subsampling*. In node or graph subsampling, we aim to reduce the number of nodes or graphs, respectively, in a graph dataset while preserving performance of a downstream GNN trained on the dataset. Existing work on node and graph subsampling focuses primarily on node-level tasks (e.g., node classification), and often makes strong assumptions about the GNN model architecture—and can even assume access to the final model or the ability to approximate the training trajectories on the original, unsampled dataset.

In this paper, we propose and theoretically motivate new approaches for (1) subsampling nodes from each training graph and (2) subsampling entire graphs in the training set for graph-level learning tasks (e.g., graph classification and

[1]Stanford University, Stanford, CA United States [2]Massachusetts Institute of Technology, Cambridge, MA United States [3]Johns Hopkins University, Baltimore, MA United States. Correspondence to: Ishani Karmarkar <ishanik@stanford.edu>.

Accepted to the Workshop on Advancing Neural Network Training at International Conference on Machine Learning (WANT@ICML 2024).

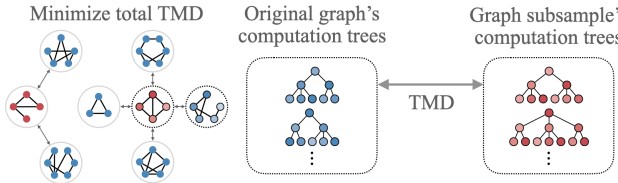

*Figure 2.* Graph subsampling. Given a graph budget $k$, the goal is to find a sample of $k$ graphs that minimizes the total distance (TMD) from every original graph to its closest sample.

regression). We must ensure that training a GNN on the subsampled data has minimal accuracy reduction compared to training on the original training set. Moreover, in practice, we often do not know *a priori* (i.e., before collecting, compressing, and labeling our graph dataset) the architecture or hyperparameters of the GNN. Consequently, it is crucial that the original and subsampled datasets maintain *model-agnostic similarity*, i.e., they produce similar model behavior regardless of the downstream model.

For many GNNs, specific *structural* similarities—tied to the GNN's computations—are essential to successfully transfer a GNN out of distribution (Yehudai et al., 2021; Chuang & Jegelka, 2022; Ruiz et al., 2021; Levie et al., 2021; Le & Jegelka, 2023). A GNN uses *message-passing* to compute predictions about graphs. Initially, each node has an embedding, which it updates by aggregating its neighbors' embeddings and processing the result through a neural network. This process repeats over $L$ iterations—or *layers*—and the nodes' final embeddings are used to compute the GNN's predictions. The $L$-hop neighbors involved in a node's final embedding can be visualized via the node's *computation tree*, illustrated in Figure 3. Computation trees play a critical role in the transfer and stability of GNNs (Yehudai et al., 2021; Chuang & Jegelka, 2022; Georgiev et al., 2022). Two nodes with similar computation trees will likely have similar embeddings. To capture this intuition, Chuang & Jegelka (2022) introduced a graph distance—the *Tree Mover's Distance (TMD)*—as an optimal transport distance between sets of computation trees, and demonstrated its correlation with the performance and stability of GNNs under data perturbations, including generalization bounds.

Based on the strong theoretical foundations provided by TMD, our primary goal is to ensure that the TMD between the original and subsampled graphs is small. However, achieving this goal requires computing the TMD between many graphs while subsampling, which poses a significant challenge. TMD is a combinatorially complex function: the dynamic programming algorithm developed in prior work for computing the TMD between graphs with $n$ nodes has a runtime of $\mathcal{O}(Ln^4)$ (Chuang & Jegelka, 2022). Given our objective to reduce the size of large graphs, this runtime is prohibitively large and presents a computational challenge

that we overcome in our methodological contributions.

**Contributions.** In this work, we make the following contributions: We present an approach for node subsampling that, given a budget $k$ and a training graph, returns a subgraph on $k$ nodes with low TMD from the original graph (Figure 1). Moreover, we present an approach for graph subsampling that, given a training set of graphs, returns $k$ graphs such that the total TMD between each original graph and its closest sampled graph is small (Figure 2).

We show that if $h$ is a GNN with minimal loss on the training set subsampled using our methods, then $h$ also has minimal loss on the original training set. When subsampling nodes in each training graph, we bound the resulting increase in loss by the average TMD between the original and subsampled graphs. When subsampling entire graphs, we bound the increase in loss by the average TMD between each training graph and the nearest graph in the subsampled set.

Given a set of candidate subgraphs, we develop a faster algorithm for selecting a subgraph with minimum TMD from a given graph. It computes the depth-$L$ TMD between a graph $G = (V, E)$ and any of its subgraphs in just $\mathcal{O}(L|E|)$ time. This algorithm leverages structural properties of the TMD that we uncover, which may be of independent interest.

Our experiments demonstrate our methods outperform or perform competitively with other standard approaches on real-world graph learning benchmarks. For graph sampling, our method consistently outperforms the baselines on four benchmark datasets, achieving over $90\%$ of the test accuracy of the full training set for three of the datasets using $1\%$ of the graphs. For node subsampling, it is consistently optimal or near-optimal across all datasets, unlike other competitive baselines, whose performance varies significantly with the fraction of deleted data.

## 2. Related work

Graph reduction, graph sparsification, and graph condensation approaches aim to to reduce the size of a graph while minimizing the impact on test loss. Existing approaches focus on node subsampling (e.g., Hashemi et al., 2024; Tanaka et al., 2020, and references therein), primarily for *node classification* and graph signal processing. These approaches involve subsampling nodes, and do not extend to graph subsampling, a critical focus of this work, and a setting which remains largely unstudied in the graph reduction literature. Two existing approaches to *model-agnostic* graph reduction for graph classification are Herding (Welling, 2009) and $k$-Centers (Farahani & Hekmatfar, 2009), which subsample $k$ nodes by clustering the nodes' features into $k$ clusters and selecting the centers. A key advantage of our approach is that, unlike these approaches, we leverage for the node features *and* the graph structure.

Gradient-matching approaches (Jin et al., 2022; Zhang et al., 2024; Fang et al., 2024) have been used for graph dataset subsampling and ensure that the loss's gradients remain similar when trained on the small and large graphs. While these are powerful methods that can aid in developing reduced graph datasets that are smaller and more interpretable, they are fundamentally incomparable in our setting because (1) they assume knowledge of a downstream GNN architecture and therefore are not *model-agnostic*; and (2) they require training on the complete, unreduced graphs to generate the gradient trajectories. This obviates several of the advantages of graph reduction with respect to graph labeling and storage costs–and these are the primary motivations for our model agnostic setting.

Georgiev et al. (2022) study a normalized variant of TMD to select training datasets for neural algorithmic reasoning tasks. However, this is not agnostic to the downstream model/task and is not directly applicable to subsampling.

## 3. Notation and background

**Notation.** An undirected, unweighted graph $G = (V, E, f)$ has a node set $V$, edges $E$, and node features $f \in \mathbb{R}^{p \times |V|}$, where $f^v$ is the feature vector of $v \in V$. We use $N_G(v)$ to denote the neighborhood of $v \in V$ and $\mathcal{G}$ to denote a set of graphs. An $r$-rooted tree is denoted by $T = (V, E, f, r)$. We use $\mathbf{1}, \mathbf{0}, \boldsymbol{I}$ to denote the all ones and zeros vectors and identity matrix, respectively. For norms, $\|\cdot\|$ denotes any norm over $\mathbb{R}^p$ and $\|\cdot\|_1$ is the $\ell_1$ norm.

**Message-passing neural networks (GNNs).** We analyze *message-passing graph neural networks (GNNs)*, which operate on graphs $G = (V, E, f)$. For simplicity, we define *Graph Isomorphism Networks (GINs)* (Xu et al., 2019) here; however, the results of Chuang & Jegelka (2022) and, therefore, our results also apply to other GNNs. A GIN is defined by $L$ (learnable) functions $\phi^{(\ell)} : \mathbb{R}^d \to \mathbb{R}^d$, such as (shallow) feed-forward neural networks, and a fixed weight $\eta > 0$. Each node begins with an initial embedding $z_v^{(0)} = f^v$ and updates its embedding using the following message-passing routine:

**Message passing layers:** for $\ell \in [L-1]$,

$$z_v^{(\ell)} = \phi^{(\ell)}\big(z_v^{(\ell-1)} + \eta \sum_{u \in N(v)} z_u^{(\ell-1)}\big),$$

**Graph readout:** $h(G) = \phi^{(L)}\big(\sum_{v \in V} z_u^{(L-1)}\big).$

Other GNN architectures may, for instance, replace the summations with an average (as in the case of Graph Convolutional Neural Networks (GCNs) (Kipf & Welling, 2016)) or other aggregation functions (Hamilton, 2020).

**Tree Mover's Distance (TMD).** The TMD was introduced by Chuang & Jegelka (2022) and has found applica-

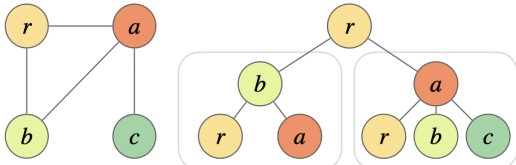

*Figure 3.* A graph and the depth-2 computation tree $T$ of node $r$. The set $\mathcal{T}_r(T)$ consists of the two boxed subtrees.

tions for graph learning under distribution shifts (Yu et al., 2023; Georgiev et al., 2022) and adversarially robust graph learning (Schuchardt et al., 2024). TMD is a pseudometric on graphs based on the optimal transport distance. Like a message passing GNN, it views a graph as a set of *computation trees*. A node's computation tree is constructed by adding the node's neighbors to the tree level by level, as in Figures 1 and 3.

**Definition 3.1** (Computation tree). Given $G = (V, E, f)$ and a node $v \in V$, let $T_v^1(G) = (v, \emptyset, \{f^v\}, v)$ be a singleton $v$-rooted tree that consists of only the node $v$, no edges, and the node feature vector $f^v$. Inductively, we define $T_v^L(G)$ to be the depth-$L$ computation tree of node $v$. That is, $T_v^L(G)$ is a tree rooted at $v$ obtained as follows. For each leaf $\ell$ of $T_v^{L-1}(G)$ and each neighbor $u \in N_G(\ell)$, we add a new node $\ell_u$ to $T_v^L(G)$ and add an edge from $\ell$ to $\ell_u$. The multiset of depth-$L$ computation trees defined by $G$ is denoted by $\mathcal{T}_G^L := \{T_v^L(G)\}_{v \in V}$.

We can compare graphs by comparing their computation trees. TMD does this via a hierarchical optimal transport.

**Definition 3.2** (OT distance). Let $X = \{x_i\}_{i=1}^q$ and $Y = \{y_i\}_{i=1}^q$ be two multisets of $q$ elements each. Given a distance metric $d : X \times Y \to \mathbb{R}$, let $C \in \mathbb{R}^{q \times q}$ be a matrix where $C_{i,j} = d(x_i, y_j)$ is the *transportation cost* between $x_i$ and $y_j$. Moreover, let the set of all *transportation plans* between $X$ and $Y$ be $\Gamma(X, Y) := \{\gamma \in \mathbb{R}_+^{q \times q} \mid \gamma \mathbf{1} = \gamma^\top \mathbf{1} = \mathbf{1}\}$. The *transportation cost* of $\gamma \in \Gamma$ is $\langle \gamma, C \rangle := \sum_{i,j} \gamma_{i,j} C_{i,j}$. The (unnormalized) Wasserstein distance $\mathrm{OT}_d$ is defined as $\mathrm{OT}_d(X, Y) := q \min_{\gamma \in \Gamma(X,Y)} \langle C, \gamma \rangle$. We say $\gamma^\star$ *is an optimal transport (OT) plan* if $\gamma^\star \in \mathrm{argmin}_{\gamma \in \Gamma(X,Y)} \langle C, \gamma \rangle$.

We need a metric between trees to compute the OT distance between sets of computation trees. TMD compares two trees $T_v, T_u$ by comparing their roots $f_v, f_u$ and recursively comparing their subtrees.

**Definition 3.3** (Tree multisets). Given an $r$-rooted tree $T = (V, E, f, r)$, let $\ell = \mathrm{depth}(T)$. We use $\mathcal{T}_r(T)$ to denote the multiset of depth $(\ell - 1)$ computation trees of all neighbors $u \in N_T(r)$, illustrated in Figure 3. (In other words, if $T_u$ is the subtree of $T$ rooted at $u$, then

$\mathcal{T}_r(T) := \cup_{u \in N_T(r)} T_u^{(\ell-1)}(T_u)$, where the union denotes a multiset union).

TMD compares $\mathcal{T}_u(T)$ and $\mathcal{T}_v(T')$ via an OT distance. However, the number of subtrees could differ in $\mathcal{T}_u(T)$ and $\mathcal{T}_v(T')$. Chuang & Jegelka (2022) therefore augment the smaller set with *blank trees*. A blank tree $T_0$ consists of a single node with node features $\mathbf{0} \in \mathbb{R}^d$ and no edges. Given two tree multisets $\mathcal{T}_v(T')$ and $\mathcal{T}_u(T)$, the function $\rho(\mathcal{T}_v(T'), \mathcal{T}_u(T))$ returns two multisets of the same size, where the smaller is padded with blank trees (see Appendix A.3 for more details). We are now ready to define a distance metric between trees.

**Definition 3.4** (Tree distance). Let $T_a = (V_a, E_a, f_a, r_a)$ and $T_b = (V_b, E_b, f_b, r_b)$ be two trees with $\ell := \max(\text{depth}(T_a), \text{depth}(T_b))$. Moreover, let $w : \mathbb{Z}_{\geq 0} \to \mathbb{R}$ be a depth-dependent weighting function. The *tree distance* (TD) between $T_a$ and $T_b$ is defined as

$$\text{TD}_w(T_a, T_b) := \|f_a^{r_a} - f_b^{r_b}\|$$
$$+ \begin{cases} w(\ell) \cdot \text{OT}_{\text{TD}_w}(\rho(\mathcal{T}_{r_a}(T_a), \mathcal{T}_{r_b}(T_b))), & \text{if } \ell > 1 \\ 0, & \text{otherwise.} \end{cases}$$

Here, $\text{OT}_{\text{TD}_w}$ is the OT distance with metric $\text{TD}_w$. We define the TMD, which calculates the cost of the optimal transport plan between two graphs' computation trees.

**Definition 3.5** (Tree mover's distance (TMD) (Chuang & Jegelka, 2022)). Given graphs $G$ and $G'$, weight function $w : \mathbb{Z}_{\geq 0} \to \mathbb{R}$, and a depth parameter $L > 0$, we define $\text{TMD}_w^L(G, G') := \text{OT}_{\text{TD}_w}(\rho(\mathcal{T}_G^L, \mathcal{T}_{G'}^L))$. The *tree norm* is $\|G\|_{\text{TN}_w^L} := \text{TMD}_w^L(G, \emptyset)$ where we overload $\emptyset$ to denote the *empty graph*.

The tree distance (Definition 3.4) and TMD (Definition 3.5) is defined *recursively*, so is intractable to compute naively. However, as noted by Chuang & Jegelka (2022), it can be computed in polynomial time using dynamic programming.

## 4. Node subsampling

In this section, we present an approach to subsampling the nodes of a graph dataset while preserving the performance of a downstream GNN trained on the subsampled dataset. Full proofs for all results in this section can be found in Appendix C. To this end, suppose we are given a graph dataset $X = \{G_1, ..., G_n\}$ where $G_i = (V_i, E_i)$ together with a node budget $k$. Given a subset of $k$ nodes $S_i \subseteq V_i$, we use $G_i[S_i]$ to denote the subgraph of $G_i$ induced by $S_i$. Our goal is to select these subsets so that a GNN obtains similar readouts on the original training dataset $X$ and on the induced subgraphs $X' = \{G_1[S_1], ..., G_n[S_n]\}$. Although quantifying the stability of the GNN readouts with respect to node deletion directly is analytically intractable,

it is possible to bound the Lipschitz constant of a message-passing GNN's readouts in terms of the TMD.[1]

**Theorem 4.1** (Informal, Theorem 8 in (Chuang & Jegelka, 2022), restated). *There exists a weight function $w$ such that given an $(L-1)$-layer message-passing GNN with readout function $h : \mathcal{G} \mapsto \mathbb{R}^d$ and $\ell$-th layer Lipschitz constants $\phi_\ell$, for any two graphs $G_a, G_b$, $\|h(G_a) - h(G_b)\| \leq \left( \prod_{\ell=1}^{L-1} \phi_\ell \right) \cdot \text{TMD}_w^L(G_a, G_b)$.*

Theorem 4.1 implies that if the Lipschitz constants of the GNN layers and the distances $\text{TMD}(G_i[S_i], G_i)$ are small, then minimizing a Lipschitz loss over the subsampled training set $X'$ yields a nearly optimal hypothesis over $X$.[2] The following corollary summarizes this observation.

**Corollary 4.2.** *Let $\mathcal{H}$ be a hypothesis class of $(L-1)$-layer GNNs $h : \mathcal{G} \to \mathbb{R}^d$ with $\ell$-th layer Lipschitz constant upper-bounded by $\Phi_\ell$. Let $\mathcal{L} : \mathbb{R}^d \to \mathbb{R}$ be an $M$-Lipschitz loss function. Let $X = (G_1, ..., G_n)$ and $(y_1, ..., y_n)$ be a set of (training) graphs and their labels. For each $i \in [n]$, let $G_i = (V_i, E_i)$ and $S_i \subset V_i$. Suppose that $M \left( \prod_{\ell \in [L-1]} \Phi_\ell \right) \leq c$ and $\text{TMD}_w^L(G_i, G_i[S_i]) \leq \varepsilon_i$ for all $i \in [n]$. Finally, let $\hat{h} \in \mathcal{H}$ be a hypothesis with minimum loss over the subsampled training set:*

$$\hat{h} = \text{argmin}_{h \in \mathcal{H}} \sum_{i \in [n]} \mathcal{L}(h(G_i[S_i]); y_i).$$

*Then $\hat{h}$ has nearly optimal loss over the original training set as well:*

$$\frac{1}{n} \sum_{i \in [n]} \mathcal{L}\left( \hat{h}(G_i); y_i \right) \leq \min_{h \in \mathcal{H}} \frac{1}{n} \sum_{i \in [n]} \mathcal{L}(h(G_i); y_i)$$
$$+ 2c/n \cdot \sum_{i \in [n]} \varepsilon_i.$$

Corollary 4.2 illustrates that the loss incurred by training on $X'$ as opposed to $X$ is bounded by an error proportional to the average of the distances $\text{TMD}_w^L(G_i, G_i[S_i])$. Motivated by this bound, our ideal goal would be to compute the optimal solution to the following optimization problem:

$$\min_{S \subset V : |S| \leq k} \text{TMD}_w^L(G, G[S]). \tag{1}$$

We face two key challenges in computing (approximately) optimal solutions to Equation (1).

**Challenge 1: exponentially large feasible set.** The first challenge is that the number of candidate subsets $\{S \subset V : |S| \leq k\}$ grows exponentially with $k$. Therefore, we restrict Equation (1) to an appropriately chosen feasible set $\mathcal{S}$. We discuss a natural heuristic for selecting $\mathcal{S}$ in Section 4.3.

---

[1] Theorem 8 as stated in (Chuang & Jegelka, 2022) is specific to GINs, but Chuang & Jegelka (2022) generalize it to other GNNs.

[2] See, e.g., Section 5 of Chuang & Jegelka (2022) for an empirical analysis of the tightness of these Lipschitz bounds.

**Definition 4.3** (Relaxed node subsampling problem). Let $G = (V, E, f)$ be a graph, $k$ be a node budget, and $\mathcal{S} \subset \{S \subset V : |S| \leq k\}$ be a set of candidate node subsets. The *relaxed node subsampling problem* is defined as: $\min_{S \in \mathcal{S}} \text{TMD}_w^L(G, G[S])$.

**Challenge 2: computing** $\text{TMD}_w^L(G, G[S])$ **is computationally expensive.** Unfortunately, even this relaxed node subsampling problem is computationally expensive, as the TMD is expensive to compute even for a single candidate $S \in \mathcal{S}$. The original dynamic-programming algorithm for computing $\text{TMD}_w^L(G, G[S])$ requires $\mathcal{O}(L|V|^4)$-time in general (Chuang & Jegelka, 2022) (regardless of $|S|$), bringing the total runtime to $\mathcal{O}(|V|^4 L |\mathcal{S}|)$. Since we aim to reduce the size of large graphs, this runtime is likely prohibitive in practice. To address this challenge, we prove that, surprisingly, Definition 4.3 is equivalent to a much simpler optimization problem which *can* be solved efficiently.

**Theorem 4.4.** *Let* $G = (V, E, f)$ *be a graph,* $k$ *be a node budget, and* $\mathcal{S} \subset \{S \subset V : |S| \leq k\}$. *Then* $S^\star$ *is an optimal solution to the* relaxed node subsampling problem *if and only if* $S^\star \in \text{argmax}_{S \in \mathcal{S}; |S| = k} \|G[S]\|_{\text{TN}_w^L}$.

We prove Theorem 4.4 in Section 4.1. Along the way, we prove several structural properties of the TMD, which may be of independent interest. Finally, since the algorithm of Chuang & Jegelka (2022) would take $\mathcal{O}(L |S|^4)$-time to compute the tree norm of $G[S]$, we provide a new linear-time dynamic programming algorithm for computing tree norms—Algorithm 1—that better leverages the graph's sparsity and runs in time only $\mathcal{O}(|E|L)$-time.

**Theorem 4.5.** *Given a graph* $G = (V, E, f)$*, Algorithm 1 computes* $\|G\|_{\text{TN}_w^L}$ *in* $O(|E| L)$*-time.*

We discuss the algorithm and proof sketch in Section 4.2, but first point out that this immediately implies a more tractable algorithm for the relaxed node subsampling problem!

**Remark 4.6.** There is an algorithm to solves Definition 4.3 in $O(k^2 |\mathcal{S}|)$ time by computing $\|G[S]\|_{\text{TN}_w^L}$ for all $S \subset \mathcal{S}$ and outputting $S$ with largest tree norm.

### 4.1. Analysis of TMD between graphs and subgraphs

In this section, we sketch our proof of Theorem 4.4. The proofs in this section are inductive in order to leverage the recursive structure of the TMD (Definition 3.5) and tree distance (Definition 3.4.) However, we emphasize that our final algorithmic result Algorithm 1 is a pure dynamic programming algorithm; i.e., the recursive formulation of TMD is useful for the analysis but is circumvented via dynamic programming in our proposed implementation.

Along the way towards proving Theorem 4.4, we prove several properties of the TMD, which may be of independent

interest given the broad applications of TMD to out-of-distribution generalization. Our analysis crucially relies on the following technical lemma.

**Lemma 4.1.** *Let* $w : \mathbb{Z}_{\geq 0} \to \mathbb{R}$ *be a weight function and* $G = (V, E, f)$ *a graph. Then, the identity transportation plan* $\mathbf{I}$ *mapping* $T_v(G)$ *to* $T_v(G[S])$ *for* $v \in S$, *and* $T_u(G)$ *to* $\emptyset$ *for* $u \notin S$ *is an optimal plan for the OT problem*

$$\text{TMD}_w^L(G, G[S]) = \text{OT}_{\text{TD}_w} \left( \rho \left( \mathcal{T}_G^L, \mathcal{T}_{G[S]}^L \right) \right). \quad (2)$$

*Consequently, we can decompose the right-hand-side of* (2) *into three terms: a sum over the feature norms of deleted nodes* $v \notin S$*; the cost of transporting the deleted nodes' computation trees (that is,* $\{\mathcal{T}_v(T_v^L(G))\}_{v \notin S}$*) to the empty set; and the cost of transporting the computation trees of the remaining nodes* $v \in S$ *(that is,* $\{\mathcal{T}_v(T_v^L(G))\}_{v \in S}$*) to their computation trees in* $G[S]$*. Formally,*

$$\text{OT}_{\text{TD}_w} \left( \rho \left( \mathcal{T}_G^L, \mathcal{T}_{G[S]}^L \right) \right) = \sum_{v \notin S} \|f^v\|$$
$$+ w(L-1) \sum_{v \notin S} \text{OT}_{\text{TD}_w} \left( \rho \left( \mathcal{T}_v \left( T_v^L(G) \right), \emptyset \right) \right)$$
$$+ w(L-1) \sum_{v \in S} \text{OT}_{\text{TD}_w} \left( \rho \left( \mathcal{T}_v \left( T_v^L(G) \right), \mathcal{T}_v \left( T_v^L(G[S]) \right) \right) \right).$$

*Proof sketch.* We prove this by induction on $L$. In the base case ($L = 1$), the statement is equivalent to Definition 3.5. This is because the definition of $\rho$ and the tree distance TD imply that $\text{OT}_{\text{TD}_w}(\rho(\mathcal{T}_G^1, \mathcal{T}_{G[S]}^1))$ is equal to the sum of features of the nodes that are in $G$ but not in $G[S]$. That is, $\text{OT}_{\text{TD}_w}(\rho(\mathcal{T}_G^1, \mathcal{T}_{G[S]}^1)) = \sum_{v \notin S} \|f^v\| = \langle C, \mathbf{I} \rangle$. This shows the lemma holds in the base case. We prove the inductive step by leveraging recursive formulation of the TD to reduce OT problems of depth $L$ to OT problems of depth $L-1$, and then applying the inductive hypothesis. ∎

Lemma 4.1 implies the following chain rule for the TMD induced by dropping a sequence of nodes from the graph.

**Lemma 4.2.** *For* $T \subset S \subset V$*,* $\text{TMD}_w^L(G, G[S \setminus T]) = \text{TMD}_w^L(G, G[S]) + \text{TMD}_w^L(G[S], G[S \setminus T])$.

*Proof sketch.* To prove the lemma, we apply Lemma 4.1 inductively over $L$ to decompose $\text{TMD}_w^L(G, G[S \setminus T])$ into $\text{TMD}_w^L(G, G[S])$ and $\text{TMD}_w^L(G[S], G[S \setminus T])$. To highlight the intuition, we discuss the base case here and discuss the inductive step in the full proof. If $L = 1$, then the second and third terms in the expression of Lemma 4.1 are always 0, because $\mathcal{T}_G^1$ and $\mathcal{T}_{G[S]}^1$ are multisets of depth 1, so $\mathcal{T}_G(T_v^L(G))$ and $\mathcal{T}_{G[S]}(T_v^L(G[S]))$ are empty sets. Thus,

$$\text{TMD}_w^1(G, G[S \setminus T]) = \sum_{v \notin (S \setminus T)} \|f^v\|$$
$$= \sum_{v \notin S} \|f^v\| + \sum_{v \in T} \|f^v\|,$$

---

**Algorithm 1:** TreeNorm$(G, L, w)$

**Input:** Graph $G = (V, E, f)$ with adjacency matrix
$A$, weights $w : \{1, ..., L-1\} \to \mathbb{R}_+, L \geq 1$

1 Compute $x \in \mathbb{R}^{|V|}$ to be the vector such that
$x_v = \|f^v\|$ ;

2 Initialize $z^{(0)} = x$;

3 **for** $\ell \in [L-1]$ **do**

4 $\quad \lfloor \quad z^{(\ell)} \leftarrow Az^{(\ell-1)}$;

5 $b \leftarrow z^{(0)} + \sum_{\ell=1}^{L-1} \left( \prod_{t=1}^{\ell} w(L-t) \right) \cdot z^{(\ell)}$;

**return:** $\|b\|_1$

---

(since $T \subset S$, $\{v \in V : v \notin (S \backslash T)\} = \{v \in V : v \notin S\} \sqcup \{v \in T\}$). Applying Lemma 4.1 to $\text{TMD}_w^L(G, G[S \backslash T])$ and $\text{TMD}_w^1(G[S], G[S \backslash T])$ implies that the base case holds:

$$\text{TMD}_w^1(G, G[S]) = \sum_{v \notin S} \|f^v\|, \text{ and}$$

$$\text{TMD}_w^1(G[S], G[S \backslash T]) = \sum_{v \in T} \|f^v\|.$$

The $L > 1$ case is similar but requires care to handle the recursive OT terms. ∎

We briefly point out why Lemma 4.2 is surprising. TMD is a pseudo-metric on graphs; so, in general, we should expect the following triangle *inequality* $\text{TMD}_w^L(G, G[S \backslash T]) \leq \text{TMD}_w^L(G, G[S]) + \text{TMD}_w^L(G[S], G[S \backslash T])$ to hold. However, Lemma 4.1 says that $T \subset S \subset V$ is enough to ensure that the inequality is *in fact* an *equality*. Intuitively, this means that TMD is additive along any sequence of vertex-induced subgraphs that reduces the full graph $G$ to the empty graph. We use this observation to prove Theorem 4.4.

*Proof of Theorem 4.4.* Let $S \in \mathcal{S}$. By Lemma 4.2 with $T = V \backslash S$, we have that $\|G\|_{\text{TN}_w^L} = \text{TMD}_w^L(G, G[S]) + \|G[S]\|_{\text{TN}_w^L}$. Consequently, $\max_{S \in \mathcal{S}; |S|=k}$ is equivalent to $\max_{S \in \mathcal{S}; |S|=k} \|G\|_{\text{TN}_w^L} - \text{TMD}_w^L(G, G[S])$, which is in turn equivalent to $\min_{S \in \mathcal{S}; |S|=k} \text{TMD}_w^L(G, G[S])$. ∎

### 4.2. Faster algorithm for computing tree norms

We next leverage Lemma 4.1 to obtain Algorithm 1, a faster algorithm for computing $\|G\|_{\text{TN}_w^L}$ for $G = (V, E, f)$. The original implementation would require $O(L|V|^4)$ time (Chuang & Jegelka, 2022), whereas ours has runtime $O(L|E|)$. Algorithm 1 is based on the fact that by Lemma 4.1, is that $\|G\|_{\text{TN}_w^L}$ is essentially a weighted sum of the number of vertices in all of its $L$-hop computation trees. For each tree, nodes in level $\ell$ receive weight $w(L - \ell + 1)$. Algorithm 1 computes this weighted sum. Its runtime is dominated by the cost of $L$ matrix-vector multiplies with the adjacency matrix, which requires $O(|E|)$-time. The proof of correctness and runtime (Theorem 4.5) is in Appendix C.

### 4.3. Heuristic for selecting $\mathcal{S}$

We now propose our heuristic for constructing the set of candidate subsets $\mathcal{S}$ in our experiments. Most real-world networks are scale-free and small-world, and hence can be well-modeled by random graphs, e.g., preferential attachment, configuration models, and inhomogeneous random graphs (Bollobás & Riordan, 2004; Newman & Watts, 1999). These graphs converge *locally* to graph limits (Van Der Hofstad, 2024). By a "transitive" argument, we can view real-world networks as parts of sequences converging to graph limits in the same sense. Alimohammadi et al. (2023) showed that for such graphs, sampling nodes' local BFS trees (Definition 4.7) asymptotically preserves motifs of the graph limit (see Appendix A).

**Definition 4.7** ($k$-BFS subset). Given $G = (V, E, f)$ and $v \in V$, let $\ell_k$ be the deepest level such that the $v$-rooted BFS tree of depth $\ell_k$ has at most $k$ nodes (breaking ties in a fixed but arbitrary way). The $k$-BFS subset of $v$, denoted $S_{\text{BFS}(v;k)}$, is the set of nodes at distance at most $\ell_k$ from $v$.

We define the set of candidate subsets $\mathcal{S} = \{S_{\text{BFS}(v;k)}\}_{v \in V}$ as the nodes' $k$-BFS subsets. This definition of $\mathcal{S}$ is model-agnostic as it is determined solely by the graph structure and is independent of the downstream mode architecture or hyperparameters. In sufficiently sparse graphs $\mathcal{S}$ is efficient to compute; however, in denser graphs, there are several previously studied approaches for subsampling the $k$-BFS subsets for improved efficiency (see, e.g. (Alimohammadi et al., 2023) and references therein) for an overview.

## 5. Graph subsampling

We now present an approach that uses TMD to subsample the number of graphs in a dataset while preserving the performance of a downstream GNN trained on the subsampled dataset. Omitted proofs are in Section C. Let $X = \{G_1, ..., G_n\}$ be a graph dataset. Given a budget $k$, the goal is to identify a subset $\mathcal{I} \subset [n]$ of size $k$ such that a GNN obtains similar readouts and loss on the original training set $X$ and on the subsampled set $\{G_i\}_{i \in \mathcal{I}}$. We must ensure the selected set $\mathcal{I}$ is representative of the full dataset. We propose selecting $\mathcal{I}$ to minimize the medoids objective.

**Definition 5.1** (Medoids objective). Let $X = \{G_1, ..., G_n\}$ be a graph dataset and $\mathcal{I} \subset [n]$ with $|\mathcal{I}| \leq k$. The medoids objective value of $\mathcal{I}$ on $X$ with respect to $\text{TMD}_w^L$ is defined

$$f_{\text{TMD}_w^L}(\mathcal{I}; X) = 1/|X| \cdot \sum_{i \in [n]} \min_{j \in \mathcal{I}} \text{TMD}_w^L(G_i, G_j).$$

For $j \in \mathcal{I}$, let $\tau_j$ be the number of graphs in $X$ closest to $G_j$ (the $j$-th medoid's cluster size): $\tau_j := \left| \{i : \text{TMD}_w^L(G_i, G_j) < \text{TMD}_w^L(G_i, G_k), \forall k \neq j \in \mathcal{I}\} \right|$, breaking ties arbitrarily.

The following lemma shows how we can bound the differ-

ence in a GNN's loss on the subsampled dataset $\mathcal{I} \subset X$ and the original dataset $X$ in terms of this medoids objective.

**Lemma 5.1.** *Let $\mathcal{H}$ be a hypothesis class of $(L-1)$-layer GNNs $h : \mathcal{G} \to \mathbb{R}^d$ with $\ell$-th layer Lipschitz constant upper-bounded by $\Phi_\ell$. Let $\mathcal{L} : \mathbb{R}^d \to \mathbb{R}$ be an $M$-Lipschitz loss function. Let $y = (y_1, ..., y_n)$ be labels for $X$ and $\mathcal{I}$ be a subset of $[n]$. Then for any GNN $h \in \mathcal{H}$ and $G \in \mathcal{G}$,*

$$\frac{1}{n} \Big| \sum_{i \in \mathcal{I}} \tau_i \mathcal{L}(h(G_i); y_i) - \sum_{i \in [n]} \mathcal{L}(h(G_i); y_i) \Big| \leq$$
$$M \cdot \prod_{\ell \in [L-1]} \Phi_\ell \cdot f_{\mathrm{TMD}_w^L}(\mathcal{I}; X).$$

*Proof sketch.* We use the Lipschitzness and Theorem 4.1 to bound the average deviation in the loss on $X$ and on $\mathcal{I}$. ∎

Lemma 5.1 implies that if $M \cdot \prod_{\ell \in [L-1]} \Phi_\ell \cdot f_{\mathrm{TMD}_w^L}(\mathcal{I}; X)$ is small and we minimize loss over the subsampled training set, we obtain a hypothesis with nearly optimal loss over the entire training set. We summarize this observation below.

**Corollary 5.2.** *Suppose* $M \left( \prod_{\ell \in [L-1]} \Phi_\ell \right) \leq c$ *and* $f_{\mathrm{TMD}_w^L}(\mathcal{I}; X) \leq \varepsilon$. *Let $\hat{h} \in \mathcal{H}$ be the hypothesis minimizing loss on the subsampled graphs $\sum_{i \in \mathcal{I}} \tau_i \mathcal{L}(h(G_i); y_i)$. Then*

$$\frac{1}{n} \sum_{i \in [n]} \mathcal{L}\big(\hat{h}(G_i); y_i\big) \leq \min_{h \in \mathcal{H}} \frac{1}{n} \sum_{i \in [n]} \mathcal{L}(h(G_i); y_i) + 2c\varepsilon.$$

Corollary 5.2 illustrates that the additional loss incurred by training on the subsampled graphs as opposed to $X$ is bounded by an error proportional to $f_{\mathrm{TMD}_w^L}(\mathcal{I}; X)$. This motivates us to formulate the graph subsampling problem as the problem of computing $\mathcal{I}$ that minimizes $f_{\mathrm{TMD}_w^L}(\mathcal{I}; X)$.

**Definition 5.3** (Graph subsampling problem). Let $X = \{G_1, ..., G_n\}$ be a set of graphs and $k$ a graph budget. In the *graph subsampling problem*, we must select a $k$-subset $\mathcal{I} \subset X$ that minimizes the $k$-medoids objective $f_{\mathrm{TMD}_w^L}(\mathcal{I}; X)$.

$k$-medoids is NP-hard (Kazakovtsev & Rozhnov, 2020), but there are efficient approximations, e.g., in Python (Buitinck et al., 2013). Consequently, the main computational bottleneck is the computation of TMDs between pairs of graphs $G_1, G_2 \in X$, which may be large if the graphs have *many nodes*. However, these computations can be sped up in practice, by combining graph subsampling with our node subsampling approach (see Section 4 and Remark 4.6), which is computationally efficient. See Appendix B for a detailed discussion and experiments.

## 6. Experiments

We evaluate our sampling methods on four graph benchmark datasets from the TUDataset graph benchmark library

(Morris et al., 2020). All GNNs were trained for 50 epochs with an 80%-train set and 20%-test split on CPUs. For MUTAG, COX2, and BZX we used a 3-layer Graph Isomorphism Network (Xu et al., 2019). For IMDB-BINARY we used a 3-layer Graph Convolutional Neural Network (GCN) (Kipf & Welling, 2016). Hyper-parameter details are in Appendix A.4. In all figures, the line plots and shadows indicate the means and 95% confidence intervals across 256 random trials. Figure 4 compares our TMD-based $k$-medoids graph subsampling procedure from Section 5 against alternative methods and training on the full dataset.

We are unaware of previous work that studies graph subsampling, but we compare against two clustering approaches based on Herding (Welling, 2009) and $k$-Centers (Farahani & Hekmatfar, 2009). These approaches are typically used for node subsampling and coreset selection in other domains (see Section 2). We adapt them to graph subsampling by defining distances between graphs using three popular graph kernels: Weisfeiler Lehman kernel (WL) (Shervashidze et al., 2011), the WL optimal assignment kernel (WL-OT) (Kriege et al., 2016), graphlet sampling (GS) kernel (Borgwardt & Kriegel, 2005), and the shortest paths (SP) kernel. Figure 4 shows that our method consistently outperforms uniform random sampling, Herding (with respect to all kernels), and $k$-Centers (with respect to all kernels) on all four benchmarks. Our approach achieves over 90% of the test accuracy of the full training set for MUTAG, COX2, and BZR using just 1% of the graphs. When graph labeling is very costly, this may be especially powerful.

Figure 5 compares our TMD-based node subsampling procedure from Section 4 against several alternative benchmarks and against training on the full dataset. For node subsampling, we only need to compute tree norms, so we use Algorithm 1. We compare against model-agnostic node selection policies Herding (Welling, 2009) and $k$-Centers (Farahani & Hekmatfar, 2009), and approaches by Salha et al. (2022) and Razin et al. (2023). While other approaches exhibit significant variability across datasets and subsampling levels, our approach consistently has the strongest or near-strongest performance across all datasets, sometimes requiring as little as 10% of the original number of nodes to achieve similar test accuracy as training on the full graphs.

## 7. Discussion

We proposed new node and graph subsampling approaches for compressing graph datasets while preserving performance of downstream GNNs. Our methods leverage new theoretical analysis of the TMD to ensure the subsampled data maintains structural similarities with the original dataset, and we bound the excess loss incurred by subsampling. Our methods are enabled by an algorithm we developed for selecting subgraphs with minimal TMD from a

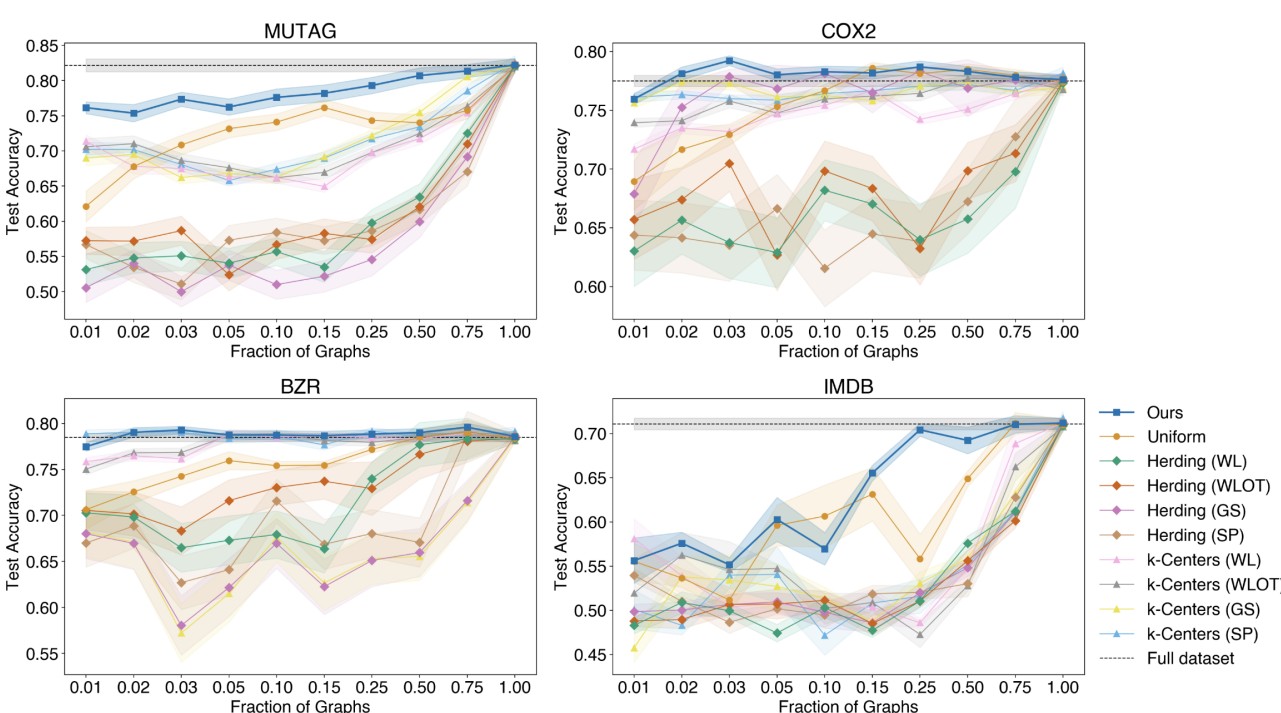

*Figure 4.* Graph subsampling. Accuracy vs. fraction of graphs in the training set, subsampled with our approach and alternative methods.

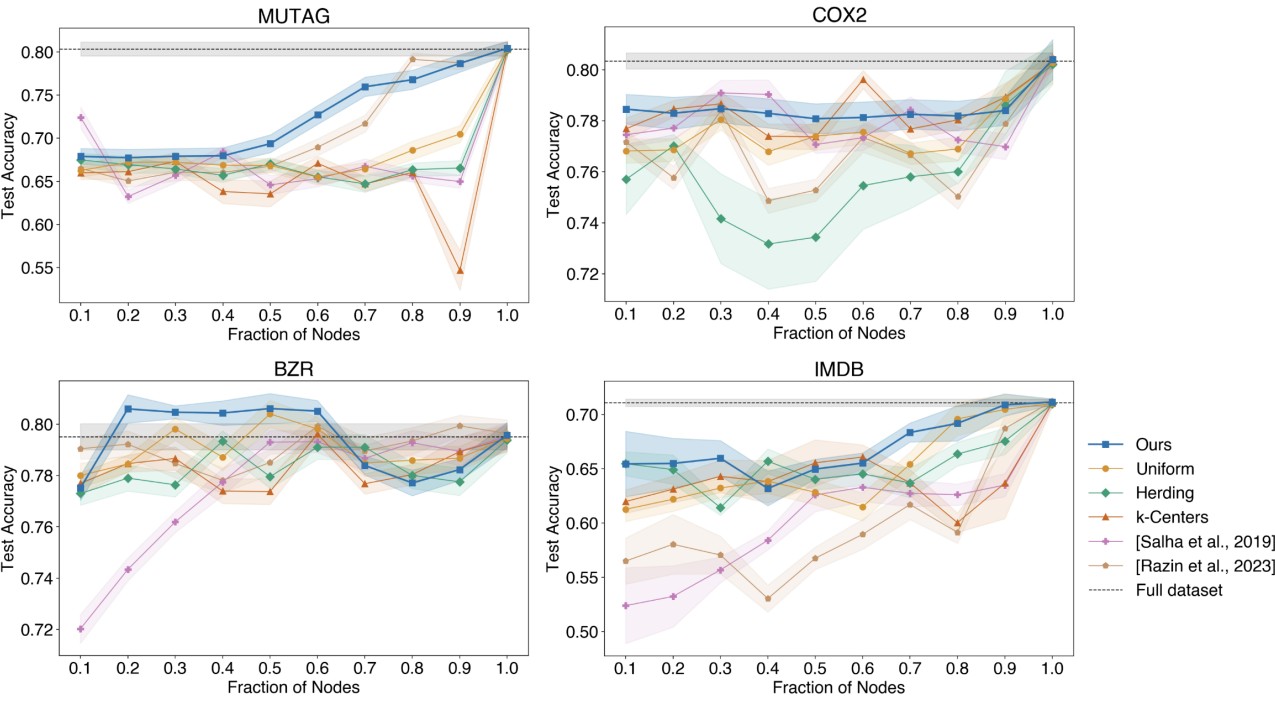

*Figure 5.* Node subsampling. Comparing accuracy vs. fraction of nodes in training graphs, subsampled with model-agnostic methods.

given graph, providing runtime speedup over prior work. Experiments demonstrate that our method outperforms other comparable model-agnostic methods for node and graph subsampling.

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

# A. Additional background

## A.1. Random graph limits

As we discussed in Section 4.3, most real networks are scale-free and small-world, and hence well-modeled by random graph models, such as preferential attachment, configuration models and inhomogenous random graphs. As shown by (Van Der Hofstad, 2024), such random graph models produce graphs that can be shown to converge *locally* to a limit $(G, o)$, where $G$ is a graph to which we assign a root node $o$. By a "transitive" argument, it makes sense to see real networks as parts of sequences converging to graph limits in a similar sense.

To be precise, let $\mathcal{G}_*$ be the set of all possible rooted graphs. A limit graph is defined as a measure over the space $\mathcal{G}_*$ with respect to the local metric

$$d_{loc}((G_1, o_1), (G_2, o_2)) = \frac{1}{1 + \inf_k \{k : B_k(G_1, o_1) \not\simeq B_k(G_2, o_2)\}}$$

where $B_k(G, v)$ is the $k$-hop neighborhood of node $v$, and $\simeq$ is the graph isomorphism. A sequence of graphs converging to this limit is defined as follows.

**Definition A.1** (Local convergence (Alimohammadi et al., 2023)). Let $G_n = (V_n, E_n)$ denote a finite connected graph. Let $(G_n, o_n)$ be the rooted graph obtained by letting $o_n \in V_n$ be chosen uniformly at random. We say that $(G_n, o_n)$ converges locally to the connected rooted graph $(G, o)$, which is a (possibly random) element of $\mathcal{G}_*$ having law $\mu$, when, for every bounded and continuous function $h : \mathcal{G}_* \to \mathbb{R}$,

$$\mathbb{E}[h(G_n, o_n)] \to \mathbb{E}_\mu(G, o)$$

where the expectation on the right-hand-side is with respect to $(G, o)$ having law $\mu$, while the expectation on the left-hand-side is with respect to the random vertex $o_n$.

## A.2. Other graph kernels

TMD is one recently proposed pseudometric on graphs, or graph kernel (Chuang & Jegelka, 2022). It is appealing in our setting because it characterizes the robustness of GNNs (see Theorem 4.1.) Other popular graph kernels include the Weisfeiler Lehman (WL) kernel (Shervashidze et al., 2011), Weisfeiler Lehman Optimal Transport (WL-OT) (Kriege et al., 2016), Shortest Paths kernel ((Borgwardt & Kriegel, 2005)), and FGW (KAWANO et al., 2024). To our knowledge, these kernels do not have comparable robustness guarantees to TMD.

## A.3. Additional details about the TMD

We use $T_0^n$ to denote $n$ disjoint copies of $T_0$. Given two tree multisets $\mathcal{T}_u(T)$, $\mathcal{T}_v(T')$, we define $\rho$ to be the following augmentation function, which returns two multi-sets of the same size:

$$\rho : (\mathcal{T}_v(T'), \mathcal{T}_u(T)) \mapsto \left( \mathcal{T}_v(T') \cup T_0^{\max(|\mathcal{T}_u(T)| - |\mathcal{T}_v(T')|, 0)}, \mathcal{T}_u(T) \cup T_0^{\max(|\mathcal{T}_v(T')| - |\mathcal{T}_u(T')|, 0)} \right).$$

## A.4. Additional details about experiments

All experiments were run on a 16-core machine with 64 GB of RAM. The GNNs were implemented using Pytorch Geometric (He et al., 2024). The TMD weight function was set according to Pascal's triangle rule (Chuang & Jegelka, 2022, Theorem 8) and our implementation of TMD was based on (Chuang & Jegelka, 2022).[3] We used the Graph Kernel Library (Siglidis et al., 2020) for the kernel distances in our experiments.

For MUTAG, COX2, and BZX we used a GIN model with the following hyperparameters:

- Number of layers: 3

- Learning rate: 0.01

- Batch size: 128

- Number of hidden channels per layer: 32

---

[3]One consideration with is the runtime required to compute the $k$-medoids. However, reduced labeling cost, reduced data storage, and privacy are often more significant priorities.

- Aggregation function: global add pool

- Weight regularization: N/A

- Optimizer: Adam

- Loss: BCE with logits

- Dropout: N/A

For IMDB we used a GCN model with the following hyperparameters:

- Number of layers: 3

- Learning rate: 0.01

- Batch size: 32

- Number of hidden channels per layer: 32

- Aggregation function: global add pool

- Weight regularization: N/A

- Optimizer: Adam

- Loss: BCE with logits

- Dropout: N/A

We refrained from using batch normalization in both models. Models were implemented using PytorchGeometric (Fey & Lenssen, 2019).

## B. Runtime speedups for graph subsampling

In this section, we show that the pairwise TMD computations required for our graph subsampling approach can be computed more efficiently using the following two-stage approach. Recall that $k$-medoids algorithms require querying distances (in our case, TMD) between two graphs $G_1, G_2$. However, if $G_1$ and $G_2$ have many nodes, computing $\text{TMD}_w^L(G_1, G_2)$ using the dynamic programming algorithm of (Chuang & Jegelka, 2022) may be computationally expensive. Fortunately, we can leverage our efficient methods for node subsampling to efficiently compute approximations of $\text{TMD}_w^L(G_1, G_2)$ as follows.

1. First, we use our node subsampling approach to create new graphs $G_1'$ and $G_2'$ such that $\text{TMD}_w^L(G_1, G_1')$ and $\text{TMD}_w^L(G_2, G_2')$ are minimized while ensuring that $G_1'$ and $G_2'$ have fewer nodes than $G_1$ and $G_2$, respectively.

2. Second, we run a $k$-medoids clustering algorithm using approximate pairwise distances $\text{TMD}_w^L(G_1', G_2')$ in place of $\text{TMD}_w^L(G_1, G_2)$. Triangle inequality ensures that

$$\left| \text{TMD}_w^L(G_1', G_2') - \text{TMD}_w^L(G_1, G_2) \right| \leq \text{TMD}_w^L(G_1, G_1') + \text{TMD}_w^L(G_2, G_2').$$

So, whenever $\text{TMD}_w^L(G_1, G_1')$ and $\text{TMD}_w^L(G_2, G_2')$ are not too large (which is the goal of our node-subsampling procedure), we know that $\text{TMD}_w^L(G_1', G_2')$ is a good approximation for $\text{TMD}_w^L(G_1, G_2)$.

To illustrate that this method can work well in practice, we conducted an experiment showing the accuracy of our $k$-medoids graph subsampling approach when using approximate TMD computations on the MUTAG dataset, as described above. For each pair of graphs $G_1, G_2$ in the MUTAG dataset, we computed $p$-approximate TMDs $\text{TMD}_w^L(G_1^p, G_2^p)$ and ran our graph subsampling approach using $\text{TMD}_w^L(G_1^p, G_2^p)$ where $G^p$ denotes the graph $G$ after using our node subsampling approach to delete a $p$-fraction of nodes in the graph. We then use $\text{TMD}_w^L(G_1^p, G_2^p)$ as a proxy for $\text{TMD}_w^L(G_1, G_2)$ in our $k$-medoids clustering step. Our results indicate that we can afford to delete a large portion (up to 70 percent) of the nodes prior to computing pairwise TMDs with only a small effect on accuracy– and computing $\text{TMD}_w^L(G_1^p, G_2^p)$ is, on average, eight-fold faster than computing $\text{TMD}_w^L(G_1, G_2)$ when $p \geq .5$. See Figure 6 for an illustration of these results.

## C. Omitted Proofs

Throughout this section, we use $\deg_G(v)$ to denote the degree of $v \in V$.

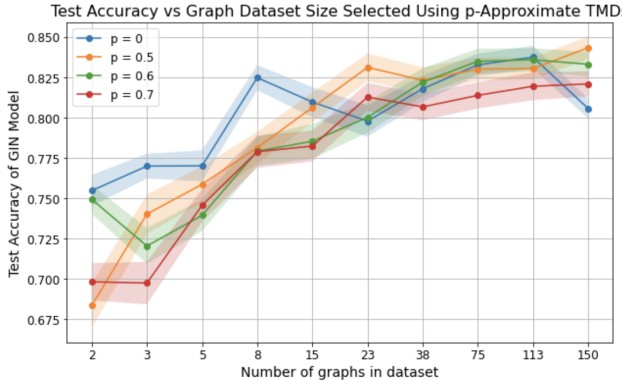

*Figure 6.* Graph subsampling with approximate TMDs. The figure shows accuracy vs number of graphs in the training dataset when sub-sampling graphs using our TMD-based graph subsampling approach with $p$-approximate TMDs.

### C.1. Omitted proofs from Section 4

**Corollary 4.2.** *Let $\mathcal{H}$ be a hypothesis class of $(L-1)$-layer GNNs $h : \mathcal{G} \to \mathbb{R}^d$ with $\ell$-th layer Lipschitz constant upper-bounded by $\Phi_\ell$. Let $\mathcal{L} : \mathbb{R}^d \to \mathbb{R}$ be an $M$-Lipschitz loss function. Let $X = (G_1, ..., G_n)$ and $(y_1, ..., y_n)$ be a set of (training) graphs and their labels. For each $i \in [n]$, let $G_i = (V_i, E_i)$ and $S_i \subset V_i$. Suppose that $M \left( \prod_{\ell \in [L-1]} \Phi_\ell \right) \leq c$ and $\mathrm{TMD}_w^L(G_i, G_i[S_i]) \leq \varepsilon_i$ for all $i \in [n]$. Finally, let $\hat{h} \in \mathcal{H}$ be a hypothesis with minimum loss over the subsampled training set:*

$$\hat{h} = \mathrm{argmin}_{h \in \mathcal{H}} \sum_{i \in [n]} \mathcal{L}(h(G_i[S_i]); y_i).$$

*Then $\hat{h}$ has nearly optimal loss over the original training set as well:*

$$\frac{1}{n} \sum_{i \in [n]} \mathcal{L}\left( \hat{h}(G_i); y_i \right) \leq \min_{h \in \mathcal{H}} \frac{1}{n} \sum_{i \in [n]} \mathcal{L}(h(G_i); y_i)$$
$$+ 2c/n \cdot \sum_{i \in [n]} \varepsilon_i.$$

*Proof.* Consider any $h \in \mathcal{H}$. By Theorem 4.1, we have that for each $i \in [n]$,

$$\|h(G_i; y_i) - h(G_i[S_i]; y_i)\| \leq \left( \prod_{\ell \in [L-1]} \Phi_\ell \right) \mathrm{TMD}_w^L(G_i, G_i[S_i]). \tag{3}$$

Consequently, using the fact that $\mathcal{L}$ is $M$-Lipschitz and (3),

$$|\mathcal{L}(h(G_i); y_i) - \mathcal{L}(h(G_i[S_i]))| \leq M \|h(G_i; y_i) - h(G_i[S_i]; y_i)\|$$
$$\leq M \left( \prod_{\ell \in [L-1]} \Phi_\ell \right) \mathrm{TMD}_w^L(G_i, G_i[S_i])$$
$$= c\mathrm{TMD}_w^L(G_i, G_i[S_i]) \leq c\varepsilon_i.$$

Consequently, by the triangle inequality,

$$\left| \frac{1}{n} \sum_{i \in [n]} \mathcal{L}(h(G_i); y_i) - \frac{1}{n} \sum_{i \in [n]} \mathcal{L}(h(G_i[S_i])) \right| \leq \frac{1}{n} \sum_{i \in [n]} |\mathcal{L}(h(G_i); y_i) - \mathcal{L}(h(G_i[S_i]; y_i))|$$
$$\leq \frac{c}{n} \sum_{i \in [n]} \varepsilon_i.$$

So, we know that for any $h \in \mathcal{H}$

$$\frac{1}{n} \sum_{i \in [n]} \mathcal{L}(h(G_i); y_i) \leq \frac{1}{n} \sum_{i \in [n]} \mathcal{L}(h(G_i[S_i]); y_i) + \frac{c}{n} \sum_{i \in [n]} \varepsilon_i. \tag{4}$$

Let $h^\star$ be the hypothesis that minimizes average loss across the original dataset $X$:

$$h^\star = \underset{h \in \mathcal{H}}{\operatorname{argmin}} \frac{1}{n} \sum_{i \in [n]} \mathcal{L}(h(G_i); y_i).$$

By definition of $\hat{h}$, this means that

$$\frac{1}{n} \sum_{i \in [n]} \mathcal{L}(\hat{h}(G_i[S_i]); y_i) \leq \frac{1}{n} \sum_{i \in [n]} \mathcal{L}(h^\star(G_i[S_i]); y_i) + \frac{c}{n} \sum_{i \in [n]} \varepsilon_i.$$

Applying (4) again to $h^\star$, we have that

$$\frac{1}{n} \sum_{i \in [n]} \mathcal{L}(\hat{h}(G_i[S_i]); y_i) \leq \frac{1}{n} \sum_{i \in [n]} \mathcal{L}(h^\star(G_i); y_i) + \frac{c}{n} \sum_{i \in [n]} \varepsilon_i.$$

∎

**Lemma 4.1.** *Let $w : \mathbb{Z}_{\geq 0} \to \mathbb{R}$ be a weight function and $G = (V, E, f)$ a graph. Then, the identity transportation plan $\boldsymbol{I}$ mapping $T_v(G)$ to $T_v(G[S])$ for $v \in S$, and $T_u(G)$ to $\emptyset$ for $u \notin S$ is an optimal plan for the OT problem*

$$\mathrm{TMD}_w^L(G, G[S]) = \mathrm{OT}_{\mathrm{TD}_w}\left(\rho\left(\mathcal{T}_G^L, \mathcal{T}_{G[S]}^L\right)\right). \tag{2}$$

*Consequently, we can decompose the right-hand-side of (2) into three terms: a sum over the feature norms of deleted nodes $v \notin S$; the cost of transporting the deleted nodes' computation trees (that is, $\{\mathcal{T}_v(T_v^L(G))\}_{v \notin S}$) to the empty set; and the cost of transporting the computation trees of the remaining nodes $v \in S$ (that is, $\{\mathcal{T}_v(T_v^L(G))\}_{v \in S}$) to their computation trees in $G[S]$. Formally,*

$$\mathrm{OT}_{\mathrm{TD}_w}\left(\rho\left(\mathcal{T}_G^L, \mathcal{T}_{G[S]}^L\right)\right) = \sum_{v \notin S} \|f^v\|$$

$$+ w(L-1) \sum_{v \notin S} \mathrm{OT}_{\mathrm{TD}_w}\left(\rho\left(\mathcal{T}_v\left(T_v^L(G)\right), \emptyset\right)\right)$$

$$+ w(L-1) \sum_{v \in S} \mathrm{OT}_{\mathrm{TD}_w}\left(\rho\left(\mathcal{T}_v\left(T_v^L(G)\right), \mathcal{T}_v\left(T_v^L(G[S])\right)\right)\right).$$

*Proof.* Let $\bar{G}[S]$ be the graph obtained by augmenting $G[S]$ with node features $\boldsymbol{0}$ for each $v \in V \setminus S$. That is, $\bar{G}[S] = (V, E[S], f')$ where $f'^v = f^v$ if $v \in S$ and $\boldsymbol{0}$ otherwise By the definition of the augmentation function $\rho$, we can instead consider the following OT problem, which is equivalent to (2):

$$\mathrm{OT}_{\mathrm{TD}_w}\left(\rho(\mathcal{T}_G^L, \mathcal{T}_{\bar{G}[S]}^L)\right).$$

In the base case where $L = 1$, we see that *any* transportation plan between the nodes of $\bar{G}[S]$ and $G$ must transport the excess node feature mass $\sum_{v \notin S} \|f^v\|$ in $G$ to some nodes in $G[S]$. The transportation plan $\boldsymbol{I}$ has cost exactly equal to $\sum_{v \in V} \|f^v - f'^v\| = \sum_{v \notin S} \|f^v\|$, so $\boldsymbol{I}$ must be an optimal transportation plan.

Now, if $L > 1$, consider the OT problem 2, and let $C$ and $\Gamma$ be the distance matrix and feasible transportation plans for this OT problem. Let $\gamma \in \Gamma$ be any feasible transportation plan.

By the definition of TD, for any $i, j \in V$, transporting $\gamma_{i,j}$ mass from $i$ to $j$ incurs two costs:

(1) $\gamma_{i,j}$ incurs the cost of transporting $f^i$ to $f'^j$.

(2) $\gamma_{i,j}$ incurs the cost of taking all of the depth $(L-1)$ computation trees of all neighbors of $i$ in $T_i^L(G)$, and transporting them to the depth $(L-1)$ computation trees of all neighbors of $j$ in $T_j^L(G[S])$ (after appropriately augmenting with isolated nodes of feature $\mathbf{0}$ for each $v \notin S$.

To quantify these two costs, let us define $\bar{\mathcal{T}}_j(T_j^L(G[S]))$ to be the multiset of computation trees obtained by augmenting $\mathcal{T}_j(T_jL(G[S]))$ with isolated nodes of feature vectors $\mathbf{0}$ for each $j \notin S$. That is, let $\bar{\mathcal{T}}_j(T_j^L(G[S]))$ be defined as the second output of $\rho$ when applied to $\left(\mathcal{T}_j(T_j^L(G)), \mathcal{T}_j(T_j^L(G[S]))\right)$ :

$$\left(\mathcal{T}_j(T_j^L(G)), \bar{\mathcal{T}}_j(T_j^L(G[S]))\right) = \rho\left(\mathcal{T}_j(T_j^L(G)), \mathcal{T}_j(T_j^L(G[S]))\right).$$

We can now quantify the cost of transportation plan $\gamma$ (2) as follows. In the following equation, the first term corresponds to item (1) above and the second term corresponds to item (2) discussed above.

$$\sum_{i,j \in V} \gamma_{i,j} C_{i,j} = \sum_{i,j \in V} \gamma_{i,j} \left\| f^i - f'^j \right\| + w(L-1) \sum_{i,j \in V} \gamma_{i,j} \mathrm{OT}_{\mathrm{TD}_w}(\mathcal{T}_i(T_i^L(G)), \bar{\mathcal{T}}_j(T_j^L(G[S]))).$$

Now, we can immediately lower bound the cost $\sum_{i,j} \gamma_{i,j} C_{i,j}$ of $\gamma$ as follows:

$$\sum_{i,j \in V} \gamma_{i,j} C_{i,j} \geq \min_{\nu \in \Gamma} \sum_{i,j \in V} \nu_{i,j} \left\| f^i - f'^j \right\|$$
$$+ w(L-1) \min_{\tau \in \Gamma} \sum_{i,j \in V} \tau_{i,j} \mathrm{OT}_{\mathrm{TD}_w}(\mathcal{T}_i(T_i^L(G)), \bar{\mathcal{T}}_j(T_j^L(G[S]))).$$

For the first term, it is easy to see that $\nu = \boldsymbol{I}$ is an optimal solution, by the same argument as in the base case: any transportation plan $\nu$ must incur the cost of transporting the excess mass $\{\|f^v\|\}_{v \notin S}$ in the node features of $G$ which is not present in the node features of $S$.

For the second term, we can notice that $\mathrm{OT}_{\mathrm{TD}_w}(\mathcal{T}_i(T_i^L(G)), \bar{\mathcal{T}}_j(T_j^L(G[S])))$ is also an TMD between graphs which contain the corresponding multisets of depth $L-1$ trees! So, by the inductive hypothesis, $\tau = \boldsymbol{I}$ is an optimal solution for this OT problem as well.

Consequently, substituting $\tau = \nu = \boldsymbol{I}$, we can further simplify the lower bound to

$$\sum_{i,j \in V} \gamma_{i,j} C_{i,j} \geq \sum_{i,j \in V} \boldsymbol{I}_{i,j} \left\| f^i - f'^j \right\| + w(L-1) \sum_{i,j \in V} \boldsymbol{I}_{i,j} \mathrm{OT}_{\mathrm{TD}_w}(\mathcal{T}_i(T_i^L(G)), \bar{\mathcal{T}}_j(T_j^L(G[S])))$$
$$= \sum_{i \in V} \left\| f^i - f'^j \right\| + w(L-1) \sum_{i \in V} \mathrm{OT}_{\mathrm{TD}_w}\left(\mathcal{T}_i(T_i^L(G), \bar{\mathcal{T}}_i(T_i^L(G[S]))\right)$$
$$= \sum_{v \notin S} \|f^v\| + w(L-1) \sum_{v \notin S} \mathrm{OT}_{\mathrm{TD}_w}\left(\rho\big(\mathcal{T}_v(T_v^L(G)), \emptyset\big)\right)$$
$$+ w(L-1) \sum_{v \in S} \mathrm{OT}_{\mathrm{TD}_w}\left(\mathcal{T}_v(T_v^L(G)), \bar{\mathcal{T}}_v(T_v^L(G[S]))\right)$$
$$= \sum_{v \notin S} \|f^i\| + w(L-1) \sum_{v \notin S} \mathrm{OT}_{\mathrm{TD}_w}\left(\rho\left(\mathcal{T}_v(T_v^L(G)), \emptyset\right)\right)$$
$$+ w(L-1) \sum_{v \in S} \mathrm{OT}_{\mathrm{TD}_w}\left(\rho\left(\mathcal{T}_v(T_v^L(G)), \mathcal{T}_v(T_v^L(G[S]))\right)\right).$$

Finally, the inequality is tight, because $\gamma = \boldsymbol{I} \in \Gamma$ is clearly a feasible transportation plan. So, by induction, the lemma holds. ∎

**Remark C.1.** The terms $\mathrm{OT}_{\mathrm{TD}_w}\left(\rho\left(\mathcal{T}_v(T_v^L(G)), \mathcal{T}_v(T_v^L(G[S]))\right)\right)$ in the expression of Lemma 4.1 can themselves be viewed as the TMD between two graphs $\tilde{G}$ and $\tilde{G}'$, where $\tilde{G}$ is the graph consisting of all elements in $\mathcal{T}_v(T_v^L(G))$ as a disjoint trees, and $\tilde{G}'$ is the graph consisting of all elements in $\mathcal{T}_v(T_v^L(G[S]))$ as disjoint trees. In this sense, Lemma 4.1 precisely characterizes the recursive structure of TMD between a graph and its subgraph. We leverage this fact in later proofs.

**Lemma 4.2.** *For $T \subset S \subset V$, $\text{TMD}_w^L(G, G[S \setminus T]) = \text{TMD}_w^L(G, G[S]) + \text{TMD}_w^L(G[S], G[S \setminus T])$.*

*Proof.* Let $Z = \{z \in V : z \notin S\}$. Since $T \subset V$, $T$ and $Z$ are disjoint. So, Lemma 4.1 shows that

$$
\begin{aligned}
\text{TMD}_w^L(G, G[S \setminus T]) = & \sum_{z \in Z} \|f^z\| + w(L-1) \sum_{z \in Z} \text{OT}_{\text{TD}_w}\left(\rho\left(\mathcal{T}_z(T_z^L(G)), \emptyset\right)\right) \\
& + \sum_{t \in T} \|f^t\| + w(L-1) \sum_{t \in T} \text{OT}_{\text{TD}_w}\left(\rho\left(\mathcal{T}_t(T_t^L(G)), \emptyset\right)\right) \\
& + w(L-1) \sum_{u \in S \setminus T} \text{OT}_{\text{TD}_w}\left(\rho\left(\mathcal{T}_u(T_u^L(G)), \mathcal{T}_u(T_u^L(G[S \setminus T]))\right)\right).
\end{aligned}
$$

Now, each term in $\text{OT}_{\text{TD}_w}\left(\rho\left(\mathcal{T}_u(T_u^L(G)), \mathcal{T}_u(T_u^L(G[S \setminus T]))\right)\right)$ inside the summation is, in fact a depth $L - 1$ TMD between two graphs corresponding to the two disjoint forests of computation trees (as we discussed in Remark C.1). Thus, by the inductive hypothesis, we can expand the last term as follows:

$$
\begin{aligned}
\sum_{u \in S \setminus T} \text{OT}_{\text{TD}_w}\left(\rho\left(\mathcal{T}_u(T_u^L(G)), \mathcal{T}_u(T_u^L(G[S \setminus T]))\right)\right) = & \sum_{u \in S \setminus T} \text{OT}_{\text{TD}_w}\left(\rho\left(\mathcal{T}_u(T_u^L(G)), \mathcal{T}_u(T_u^L(G[S]))\right)\right) \\
& + \text{OT}_{\text{TD}_w}\left(\rho\left(\mathcal{T}_u(T_u^L(G[S])), \mathcal{T}_u(T_u^L(G[S \setminus T]))\right)\right).
\end{aligned}
$$

Consequently, we obtain:

$$
\begin{aligned}
\text{TMD}_w^L(G, G[S \setminus T]) = & \sum_{z \in Z} \|f^z\| + w(L-1) \sum_{z \in Z} \text{OT}_{\text{TD}_w}\left(\rho\left(\mathcal{T}_z(T_z^L(G)), \emptyset\right)\right) \\
& + \sum_{t \in T} \|f^t\| + w(L-1) \sum_{t \in T} \text{OT}_{\text{TD}_w}\left(\rho\left(\mathcal{T}_t(T_t^L(G)), \emptyset\right)\right) \\
& + w(L-1) \sum_{u \in S \setminus T} \text{OT}_{\text{TD}_w}\left(\rho\left(\mathcal{T}_u(T_u^L(G)), \mathcal{T}_u(T_u^L(G[S]))\right)\right) \\
& + w(L-1) \sum_{u \in S \setminus T} \text{OT}_{\text{TD}_w}\left(\rho\left(\mathcal{T}_u(T_u^L(G[S])), \mathcal{T}_u(T_u^L(G[S \setminus T]))\right)\right).
\end{aligned}
$$

Next, we can add and subtract terms as follows.

$$
\begin{aligned}
\text{TMD}_w^L(G, G[S \setminus T]) = & \sum_{z \in Z} \|f^z\| + w(L-1) \sum_{z \in Z} \text{OT}_{\text{TD}_w}\left(\rho\left(\mathcal{T}_z(T_z^L(G)), \emptyset\right)\right) \\
& + \sum_{t \in T} \|f^t\| + w(L-1) \sum_{t \in T} \text{OT}_{\text{TD}_w}\left(\rho\left(\mathcal{T}_t(T_t^L(G)), \emptyset\right)\right) \\
& + w(L-1) \sum_{u \in S} \text{OT}_{\text{TD}_w}\left(\rho\left(\mathcal{T}_u(T_u^L(G)), \mathcal{T}_u(T_u^L(G[S]))\right)\right) \\
& - w(L-1) \sum_{u \in T} \text{OT}_{\text{TD}_w}\left(\rho\left(\mathcal{T}_u(T_u^L(G)), \mathcal{T}_u(T_u^L(G[S]))\right)\right) \\
& + w(L-1) \sum_{u \in S \setminus T} \text{OT}_{\text{TD}_w}\left(\rho\left(\mathcal{T}_u(T_u^L(G[S])), \mathcal{T}_u(T_u^L(G[S \setminus T]))\right)\right).
\end{aligned}
$$

Now, $\text{OT}_{\text{TD}_w}\left(\rho\left(\mathcal{T}_u(T_u^L(G)), \mathcal{T}_u(T_u^L(G[V \setminus S]))\right)\right)$ is also a depth $L - 1$ TMD between two graphs corresponding to two disjoint forests of computation trees (again, see Remark C.1). So, we can apply the inductive hypothesis to see that for each $t \in T$,

$$
\begin{aligned}
\text{OT}_{\text{TD}_w}\left(\rho\left(\mathcal{T}_t(T_t^L(G)), \emptyset\right)\right) = & \text{OT}_{\text{TD}_w}\left(\rho\left(\mathcal{T}_t(T_t^L(G)), \mathcal{T}_t(T_t^L(G[S]))\right)\right) \\
& + \text{OT}_{\text{TD}_w}\left(\rho\left(\mathcal{T}_t(T_t^L(G[S])), \emptyset\right)\right).
\end{aligned}
$$

Thus, summing over $t \in T$, we have

$$\sum_{t \in T} \mathrm{OT_{TD}}_w \left( \rho \left( \mathcal{T}_t(T_t^L(G[S])), \emptyset \right) \right) = \sum_{t \in T} \mathrm{OT_{TD}}_w \left( \rho \left( \mathcal{T}_t(T_t^L(G)), \emptyset \right) \right)$$
$$- \sum_{t \in T} \mathrm{OT_{TD}}_w \left( \rho \left( \mathcal{T}_t(T_t^L(G)), \mathcal{T}_t(T_t^L(G[S])) \right) \right).$$

By substituting this into the equation above, we obtain

$$\mathrm{TMD}_w^L(G, G[S \setminus T]) = \sum_{z \in Z} \|f^z\| + w(L-1) \sum_{z \in Z} \mathrm{OT_{TD}}_w \left( \rho \left( \mathcal{T}_z(T_z^L(G)), \emptyset \right) \right)$$
$$+ \sum_{t \in T} \|f^t\| + w(L-1) \sum_{t \in T} \mathrm{OT_{TD}}_w \left( \rho \left( \mathcal{T}_t(T_t^L(G[S])), \emptyset \right) \right)$$
$$+ w(L-1) \sum_{u \in S} \mathrm{OT_{TD}}_w \left( \rho \left( \mathcal{T}_u(T_u^L(G)), \mathcal{T}_u(T_u^L(G[S])) \right) \right)$$
$$+ w(L-1) \sum_{u \in S \setminus T} \mathrm{OT_{TD}}_w \left( \rho \left( \mathcal{T}_u(T_u^L(G[S])), \mathcal{T}_u(T_u^L(G[S \setminus T])) \right) \right).$$

Rearranging in the form of Lemma 4.1, we obtain

$$\mathrm{TMD}_w^L(G, G[S \setminus T]) = \sum_{u \notin S} \|f^u\| + w(L-1) \sum_{u \notin S} \mathrm{OT_{TD}}_w \left( \rho \left( \mathcal{T}_u(T_u^L(G)), \mathbf{0} \right) \right)$$
$$+ w(L-1) \sum_{u \in S} \mathrm{OT_{TD}}_w \left( \rho \left( \mathcal{T}_u(T_u^L(G)), \mathcal{T}_u(T_u^L(G[S])) \right) \right)$$
$$+ \sum_{t \in T} \|f^t\| + w(L-1) \sum_{t \in T} \mathrm{OT_{TD}}_w \left( \rho \left( \mathcal{T}_t(T_t^L(G[S])), \mathbf{0} \right) \right)$$
$$+ w(L-1) \sum_{t \in S \setminus T} \mathrm{OT_{TD}}_w \left( \rho \left( \mathcal{T}_t(T_t^L(G[S])), \mathcal{T}_t(T_t^L(G[S \setminus T])) \right) \right)$$
$$= \mathrm{TMD}_w^L(G, G[S]) + \mathrm{TMD}_w^L(G[S], G[S \setminus T]).$$

∎

**Theorem 4.5.** *Given a graph $G = (V, E, f)$, Algorithm 1 computes $\|G\|_{\mathrm{TN}_w^L}$ in $O(|E|\,L)$-time.*

*Proof.* Consider Algorithm 1, TreeNorm. The runtime is immediate from the algorithm pseudocode, as matrix-vector products can be computed in $O(|E|)$. In this proof, let $A_G$ denote the adjacency matrix of a graph $G$ and let $x_G$ be the vector in $\mathbb{R}^V$ such that $x_G(v) = \|f^v\|$.

To verify the correctness, we proceed by induction. We just need to show that for all $L$, $\|G\|_{\mathrm{TN}_w^L} = \|x_G\|_1 + \left\| \sum_{\ell=1}^{L-1} \left( \prod_{t=1}^{\ell} w(L-t) \right) A_G^\ell x_G \right\|_1$, as this is the exact computation computed by the algorithm pseudocode. Note that an equivalent expression is $\|G\|_{\mathrm{TN}_w^L} = \left\| \sum_{\ell=0}^{L-1} \left( \prod_{t=1}^{\ell} w(L-t) \right) A_G^\ell x_G \right\|_1$, where the product over the emptyset is defined as 1.

If $L = 1$, then $\|G\|_{\mathrm{TN}_w^L}$ is simply the sum of its feature values, so this is trivially true.

Consider the case of a depth-$L$ tree-norm computation. Throughout this proof, all variables are positive, and consequently we can move the $\ell_1$ norm in and out of sums freely (i.e., we have a triangle *equality*.)

By taking $S = \emptyset$ in Lemma 4.1, we see that $\|G\|_{\mathrm{TN}_w^L}$ is equal to the norm of its nodes plus $w(L-1) \sum_{v \in V} \left\| \mathcal{T}_v(T_v^L(G)) \right\|_{\mathrm{TN}_w^{L-1}}$. That is,

$$\|G\|_{\mathrm{TN}_w^L} = \|x_G\|_1 + w(L-1) \sum_{v \in V} \left\| \mathcal{T}_v(T_v^L(G)) \right\|_{\mathrm{TN}_w^{L-1}}.$$

To interpret this expression, let $G'_v$ be the graph which is a forest of all the trees in $\mathcal{T}_v(T_v^L(G))$– for each $u \in N_G(v)$, we $G'_v$ will contain a depth-$(L-1)$ tree $G'_{v,u}$. Then, by inductive hypothesis.

$$\left\|\mathcal{T}_v(T_v^L(G))\right\|_{\mathrm{TN}_w^{L-1}} = \sum_{u \in N_G(v)} \left\|(T_v^L(G))\right\|_{\mathrm{TN}_w^{L-1}} = \|G'_v\|_{\mathrm{TN}_w^{L-1}} = \sum_{u \in N_G(v)} \|G'_{u,v}\|_{\mathrm{TN}_w^{L-1}}$$

$$= \sum_{u \in N_G(v)} \left\| \sum_{\ell=0}^{L-2} \left( \prod_{t=1}^{\ell} w(L-t-1) \right) A_{G_{u,v}}^{\ell} x_{G_{u,v}} \right\|_1$$

$$= \sum_{u \in N_G(v)} \sum_{\ell=0}^{L-2} \left( \prod_{t=1}^{\ell} w(L-t-1) \right) \left\| A_{G_{u,v}}^{\ell} x_{G_{u,v}} \right\|_1.$$

Letting $A_G(v,u)$ be the $(v,u)$-th entry of $A_G$, we can make the outer sum go over all vertices and add an extra adjacency matrix factor on the inner sum as follows.

$$\left\|\mathcal{T}_v(T_v^L(G))\right\|_{\mathrm{TN}_w^{L-1}} = \sum_{u \in N_G(v)} \sum_{\ell=0}^{L-2} \left( \prod_{t=1}^{\ell} w(L-t-1) \right) \left\| A_{G_{u,v}}^{\ell} x_{G_{u,v}} \right\|_1$$

$$= \left\| \sum_{u \in V} \sum_{\ell=0}^{L-2} \left( \prod_{t=1}^{\ell} w(L-t-1) \right) A_G(u,v) A_{G_{u,v}}^{\ell} x_{G_{u,v}} \right\|_1.$$

Now, by the way we constructed the trees $\mathcal{T}_v(T_v^L(G))$, $u$ has the same $(L-1)$-hop neighborhood in $G$ as it does in $G_{u,v}$. Consequently,

$$\|G\|_{\mathrm{TN}_w^L} = \|x_G\|_1 + w(L-1) \sum_{v \in V} \left\|\mathcal{T}_v(T_v^L(G))\right\|_{\mathrm{TN}_w^{L-1}}$$

$$= \|x_G\|_1 + w(L-1) \left\| \sum_{\ell=0}^{L-2} \left( \prod_{t=1}^{\ell} w(L-t-1) \right) \sum_{v \in V} \sum_{u \in V} A_G(u,v) A_{G_{u,v}}^{\ell} x_{G_{u,v}} \right\|_1$$

$$= \|x_G\|_1 + \left\| \sum_{\ell=1}^{L-2} \left( \prod_{t=1}^{\ell} w(L-t-1) \right) \sum_{v \in V} A_G^{\ell+1} x_G \right\|_1$$

$$= \|x_G\|_1 + \left\| \sum_{\ell=1}^{L-1} \left( \prod_{t=1}^{\ell} w(L-t) \right) A_G^{\ell} x_G \right\|_1,$$

completing the induction and the proof of correctness.

∎

### C.2. Omitted proofs from Section 5

**Lemma 5.1.** *Let $\mathcal{H}$ be a hypothesis class of $(L-1)$-layer GNNs $h : \mathcal{G} \to \mathbb{R}^d$ with $\ell$-th layer Lipschitz constant upper-bounded by $\Phi_\ell$. Let $\mathcal{L} : \mathbb{R}^d \to \mathbb{R}$ be an $M$-Lipschitz loss function. Let $y = (y_1, ..., y_n)$ be labels for $X$ and $\mathcal{I}$ be a subset of $[n]$. Then for any GNN $h \in \mathcal{H}$ and $G \in \mathcal{G}$,*

$$\frac{1}{n} \left| \sum_{i \in \mathcal{I}} \tau_i \mathcal{L}(h(G_i); y_i) - \sum_{i \in [n]} \mathcal{L}(h(G_i); y_i) \right| \leq$$

$$M \cdot \prod_{\ell \in [L-1]} \Phi_\ell \cdot f_{\mathrm{TMD}_w^L}(\mathcal{I}; X).$$

*Proof.* Let $\kappa : [n] \to \mathcal{I}$ map $i \in [n]$ to the index $j \in \mathcal{I}$ such that $G_j$ is the closest graph in $\{G_i\}_{i \in \mathcal{I}}$ to $G_i$. That is,

$$\kappa(i) = \operatorname*{argmin}_{j \in \mathcal{I}} \mathrm{TMD}_w^L(G_i, G_j),$$

with ties broken arbitrarily (but consistently with the mapping $\tau$.) First, we can rearrange the sums as follows

$$\left| \frac{1}{n} \sum_{i \in \mathcal{G}} \tau_i \mathcal{L}(h(G_i); y_i) - \frac{1}{n} \sum_{i \in [n]} \mathcal{L}(h(G_i); y_i) \right| = \left| \frac{1}{n} \sum_{i \in [n]} \mathcal{L}(h(G_{\kappa(i)}); y_{\kappa(i)}) - \mathcal{L}(h(G_i); y_i) \right|$$

$$\leq \frac{1}{n} \sum_{i \in [n]} \left| \mathcal{L}(h(G_{\kappa(i)}; y_{\kappa(i)})) - \mathcal{L}(h(G_i); y_i) \right|$$

$$\leq \frac{M}{n} \sum_{i \in [n]} \left\| \mathcal{L}(h(G_{\kappa(i)}); y_{\kappa(i)}) - \mathcal{L}(h(G_i); y_i) \right\|,$$

where in the last inequality, we used that $\mathcal{L}$ is $M$-lipschitz. Now, by Theorem 4.1, we have

$$\frac{M}{n} \sum_{i \in [n]} \left\| \mathcal{L}(h(G_{\kappa(i)}); y_{\kappa(i)}) - h(G_i; y_i) \right\| \leq \frac{M}{n} \left( \prod_{\ell \in [L-1]} \Phi_\ell \right) \sum_{i \in [n]} \mathrm{TMD}_w^L(G_{\kappa(i)}, G_i)$$

$$= M \left( \prod_{\ell \in [L-1]} \Phi_\ell \right) f_{\mathrm{TMD}_w^L}(S; X).$$

■

**Corollary 5.2.** *Suppose* $M \left( \prod_{\ell \in [L-1]} \Phi_\ell \right) \leq c$ *and* $f_{\mathrm{TMD}_w^L}(\mathcal{G}; X) \leq \varepsilon$. *Let* $\hat{h} \in \mathcal{H}$ *be the hypothesis minimizing loss on the subsampled graphs* $\sum_{i \in \mathcal{G}} \tau_i \mathcal{L}(h(G_i); y_i)$. *Then*

$$\frac{1}{n} \sum_{i \in [n]} \mathcal{L}\left(\hat{h}(G_i); y_i\right) \leq \min_{h \in \mathcal{H}} \frac{1}{n} \sum_{i \in [n]} \mathcal{L}(h(G_i); y_i) + 2c\varepsilon.$$

*Proof.* From Lemma 5.1, we know that

$$\frac{1}{n} \sum_{i \in [n]} \mathcal{L}\left(\hat{h}(G_i); y_i\right) \leq \frac{1}{n} \sum_{i \in S} \tau_i \mathcal{L}(\hat{h}(G_i); y_i) + c\varepsilon.$$

Let $h^*$ be the hypothesis that minimizes average loss over the original dataset $X$:

$$h^* = \operatorname*{argmin}_{h \in \mathcal{H}} \frac{1}{n} \sum_{i \in [n]} \mathcal{L}(h(G_i); y_i).$$

By definition of $\hat{h}$, this means that

$$\frac{1}{n} \sum_{i \in [n]} \mathcal{L}\left(\hat{h}(G_i); y_i\right) \leq \frac{1}{n} \sum_{i \in S} \tau_i \mathcal{L}(h^*(G_i); y_i) + c\varepsilon.$$

Applying Lemma 5.1 again, we have that

$$\frac{1}{n} \sum_{i \in [n]} \mathcal{L}\left(\hat{h}(G_i); y_i\right) \leq \frac{1}{n} \sum_{i \in [n]} \mathcal{L}\left(h^*(G_i); y_i\right) + 2c\varepsilon.$$

■

