# OpenReview forum: "Model-Agnostic Graph Dataset Compression with the Tree Mover’s Distance"
_ICML.cc/2024/Workshop/WANT — WANT@ICML 2024 Poster_

### Official Review · Reviewer_wdee · 2024-06-12
**Model-Agnostic Graph Dataset Compression with the Tree Mover’s Distance**

**Confidence:** 4

**Summary:**

This paper proposes an approach to reduce the size of large graphs while maintaining the same learning model performance. Their approach making use of the tree distance between the original graph and subsampled graphs is based on a recursive computation of the cost measuring the TMD between graphs.

**Strengths:**

- The algorithm is based on a recursive measure of the distance between graphs which helps prove theorems easily based on induction.
- Strong empirical arguments prove that the approach proposed

**Weaknesses:**

- Experiments were conducted on small graph architectures, which does not answer the scalability problem, which is a good motivation for the paper.
- I am not convinced with some induction proofs, namely the proof of lemma 4.2, using the result of lemma 4.1 requires the set Z to be defined as $\{z \in V: z \in T, Z \notin S \}$ which is itself contradictory given the fact that $T \subseteq S$, reading the proof gives me the intuition that there's an inequality instead of an equation.
- Recursively defining the cost leads to the intractability of the proposed algorithm, this point was not discussed in the paper.
- In remark 4.6, you assume knowing S in advance. Isn't this similar to the strong assumption of knowing the downstream? Also defining S by a set of k-BFS sets seems to work only for small architectures.

**Suggestions:**

- It would be interesting to add experimental results for large models to truly evaluate the proposed graph compression's performance.
- I'd suggest studying the hardness of the proposed algorithm rigorously: starting with an intractable algorithm, evaluating its hardness, and seeing the degrees of freedom we can play with to make it tractable (e.g. randomize a deterministic intractable algorithm).

---

### Official Review · Reviewer_i5fu · 2024-06-14
**Interesting work for decreasing graph dataset size for efficient GNN training**

**Confidence:** 2

**Summary:**

Authors present Graph Neural Network (GNN) *model-agnostic* approaches for *both* (1) node subsampling and (2) graph dataset subsampling, which have *bounded GNN prediction accuracy decrease*, by developing an algorithm for selecting a subgraph with minimum Tree Mover Distance (TMD). Authors prove that finding optimal node subset, i.e. finding a subset with minimal TMD, is equivalent to finding a subset with a maximum tree norm, which can be solved faster as computing the norm takes linear runime compared to polynomial runtime of computing TMD. The experiments show that the proposed method preserves the GNN accuracy better or almost better (and more consistently) with respect to fraction of subsampled training set, compared to existing methods.

**Strengths:**

The paper is well structured and presents well to the reader 1) research question/context 2) goal 3) challenges/problem 4) proposed solutions. The paper seems to stand well in the context of the workshop, as it is related to efficient training of GNNs by compressing the dataset.

Though I did not fact-checked theoretical results, as I am not familiar with the graph theory, it looks like there is a lot of work done and presented to the reader (proofs, lemmas, etc.).

The empirical accuracy preserving results are interesting and important.

**Weaknesses:**

My main concern is the experimental/comparison part of the paper. I would have liked to see empirical runtime comparison of the methods, both in terms of algorithms runtime and further GNN training runtime. Since the accuracy preserving results are positive but not *extremely* significant (for some cases it works as good as random), it would be a great point.

Beside that, though the choice of the methods to compare with is well advocated (e.g., methods are model agnostic), it is still interesting to see how big the gap is in accuracy/runtime compared to non agnostic approaches (ones that were mentioned in related work section).

**Suggestions:**

If it is possible to share as plots some empirical runtime comparison - that would be great.

---

### Meta-Review · Area_Chair_EmTP · 2024-06-17

**Recommendation:** Accept (Poster)
**Confidence:** 3

**Metareview:**

## Strengths
* The paper is well presented
* Theoretical results provide insights about simplifying computations of TMD between a graph and its subgraph
* Empirical results are interesting and important, but mostly for graph sampling

## Weaknesses
* Notation-heavy paper, it might be possible to make the presentation simpler
* Experimental results do not contain any discussion about the computing requirement for doing the
  sampling.
* Experimental results for node sampling (which represents most of the theoretical work) are not so
  convincing. It would be interesting to evaluate the Lipschitz coefficients of the considered
  networks, to assess how strong the theoretical bounds are.

The general sentiment about this paper is mixed. I recommend acceptance as a poster, provided that
authors refine their experimental evaluation.

---

### Decision · Program_Chairs · 2024-06-17

**Decision:**

Accept (Poster)

**Comment:**

We thank the authors for their time and contribution to WANT and we are pleased to share that after the reviewing process the paper has been accepted. Congratulations! We encourage the authors to consider reviewers' feedback for the improvement of the camera-ready version. We hope to see you in person at the workshop and brainstorm on efficient training research together!